# SELFISH SPARSE RNN TRAINING

## ABSTRACT

Sparse neural networks have been widely applied to reduce the necessary resource requirements to train and deploy over-parameterized deep neural networks. For inference acceleration, methods that induce sparsity from a pre-trained dense network (dense-to-sparse) work effectively. Recently, dynamic sparse training (DST) has been proposed to train sparse neural networks without pre-training a large and dense network (sparse-to-sparse), so that the training process can also be accelerated. However, previous sparse-to-sparse methods mainly focus on Multilayer Perceptron Networks (MLPs) and Convolutional Neural Networks (CNNs), failing to match the performance of dense-to-sparse methods in Recurrent Neural Networks (RNNs) setting. In this paper, we propose an approach to train sparse RNNs with a fixed parameter count in one single run, without compromising performance. During training, we allow RNN layers to have a non-uniform redistribution across cell weights for a better regularization. Further, we introduce SNT-ASGD, a variant of the averaged stochastic gradient optimizer, which significantly improves the performance of all sparse training methods for RNNs. Using these strategies, we achieve state-of-the-art sparse training results, even better than dense model results, with various types of RNNs on Penn TreeBank and Wikitext-2 datasets.

## 1 INTRODUCTION

Recurrent neural networks (RNNs) (Elman, 1990), with a variant of long short-term memory (LSTM) (Hochreiter & Schmidhuber, 1997), have been highly successful in various fields, including language modeling (Mikolov et al., 2010), machine translation (Kalchbrenner & Blunsom, 2013), question answering (Hirschman et al., 1999; Wang & Jiang, 2017), etc. As a standard task to evaluate models' ability to capture long-range context, language modeling has witnessed great progress in RNNs. Mikolov et al. (2010) demonstrated that RNNs perform much better than backoff models for language modeling. After that, various novel RNN architectures such as Recurrent Highway Networks (RHNs) (Zilly et al., 2017), Pointer Sentinel Mixture Models (Merity et al., 2017), Neural Cache Model (Grave et al., 2017), Mixture of Softmaxes (AWD-LSTM-MoS) (Yang et al., 2018), ordered neurons LSTM (ON-LSTM) (Shen et al., 2019), and effective regularization like variational dropout (Gal & Ghahramani, 2016), weight tying (Inan et al., 2017), DropConnect (Merity et al., 2018) have been proposed to significantly improve the performance of RNNs.

At the same time, as the performance of deep neural networks (DNNs) improves, the resources required to train and deploy deep models are becoming prohibitively large. To tackle this problem, various dense-to-sparse methods have been developed, including but not limited to pruning (LeCun et al., 1990; Han et al., 2015), Bayesian methods (Louizos et al., 2017a; Molchanov et al., 2017), distillation (Hinton et al., 2015), $L_1$ Regularization (Wen et al., 2018), and low-rank decomposition (Jaderberg et al., 2014). Given a pre-trained model, these methods work effectively to accelerate the inference. Recently, some dynamic sparse training (DST) approaches (Mocanu et al., 2018; Mostafa & Wang, 2019; Dettmers & Zettlemoyer, 2019; Evci et al., 2020) have been proposed to bring efficiency for both, the training phase and the inference phase by dynamically changing the sparse connectivity during training. However, previous approaches are mainly for CNNs. For RNNs, the long-term dependencies and repetitive usage of recurrent cells make them more difficult to be sparsified (Kalchbrenner et al., 2018; Evci et al., 2020). More importantly, the state-of-the-art performance achieved by RNNs on language modeling is mainly associated with the optimizer, averaged stochastic gradient descent (ASGD) (Polyak & Juditsky, 1992), which is not compatible with the existing DST approaches. The above-mentioned problems heavily limit the performance

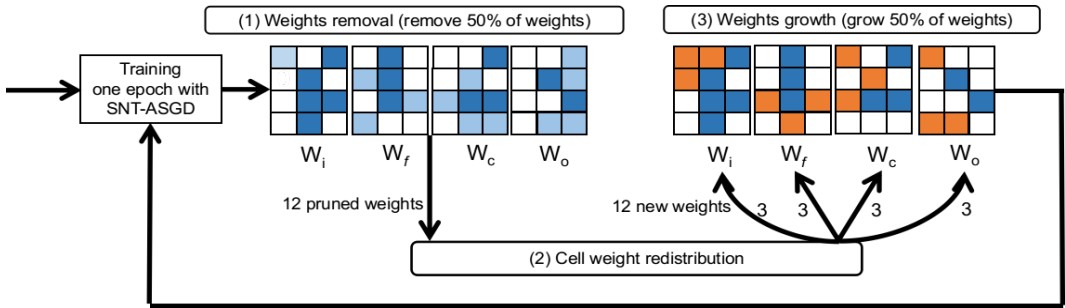

Figure 1: Schematic diagram of the Selfish-RNN. $W_i, W_f, W_c, W_o$ refer to LSTM cell weights. Colored squares and white squares refer to nonzero weights and zero weights, respectively. Light blue squares are weights to be removed and orange squares are weights to be grown.

of the off-the-shelf sparse training methods in the RNN field. For instance, while "The Rigged Lottery" (RigL) achieves state-of-the-art sparse training results with various CNNs, it fails to match the performance of the iterative pruning method in the RNN setting (Evci et al., 2020). In this paper, we introduce an algorithm to train sparse RNNs with a fixed number of computational costs throughout training. We abbreviate our sparse RNN training method as Selfish-RNN because our method encourages cell weights to obtain their parameters selfishly. The main contributions of this work are five-fold:

- We propose an algorithm to train sparse RNNs from scratch with a fixed number of parameters. This advantage constrains the training costs to a fraction of the costs needed for training a dense model, allowing us to choose suitable sparsity levels for different types of training platforms.

- We introduce SNT-ASGD, a sparse variant of the non-monotonically triggered averaged stochastic gradient descent optimizer, which overcomes the over-sparsified problem of the original NT-ASGD (Merity et al., 2018) caused by dynamic sparse training.

- We demonstrate state-of-the-art sparse training performance with various RNN models, including stacked LSTMs (Zaremba et al., 2014), RHNs, ordered neurons LSTM (ON-LSTM) on Penn TreeBank (PTB) dataset (Marcus et al., 1993) and AWD-LSTM-MoS on WikiText-2 dataset (Melis et al., 2018).

- We present an approach to analyze the evolutionary trajectory of the sparse connectivity optimized by dynamic sparse training from the perspective of graph. With this approach, we show that there exist many good structural local optima (sparse sub-networks having equally good performance) in RNNs, which can be found in an efficient and robust manner.

- Our analysis shows two surprising phenomena in the setting of RNNs contrary to CNNs: (1) random-based weight growth performs better than gradient-based weight growth, (2) uniform sparse distribution performs better than *Erdős-Rényi* (ER) sparse initialization. These results highlight the need to choose different sparse training methods for different architectures.

## 2 RELATED WORK

**Dense-to-Sparse.** There are a large amount of works operating on a dense network to yield a sparse network. We divide them into three categories based on the training cost in terms of memory and computation.

(1) *Iterative Pruning and Retraining.* To the best of our knowledge, pruning was first proposed by Janowsky (1989) and Mozer & Smolensky (1989) to yield a sparse network from a pre-trained network. Recently, Han et al. (2015) brought it back to people's attention based on the idea of iterative pruning and retraining with modern architectures. Some recent works were proposed to further reduce the number of iterative retraining e.g., Narang et al. (2017); Zhu & Gupta (2017). Frankle & Carbin (2019) proposed the Lottery Ticket Hypothesis showing that the sub-networks ("winning tickets") obtained via iterative pruning combined with their "lucky" initialization can outperform the dense networks. Zhou et al. (2019) discovered that the sign of their initialization is the crucial factor that

Table 1: Comparison of different sparsity inducing approaches. *Removal* and *Growth* refer the strategies used for adaptive sparse connectivity. *Fixed* is possible if the parameter count is fixed throughout training. *Backward Sparse* means a clear sparse backward pass and no need to compute or store any information of the non-existing weights.

| Method | Initialization | Removal | Growth | Fixed | Backward Sparse |
|---|---|---|---|---|---|
| Iterative Pruning | dense | $min(|\theta|)$ | none | no | no |
| ISS | dense | group Lasso | none | no | no |
| SET | *Erdős-Rényi* | $min(|\theta|)$ | random | yes | yes |
| DSR | uniform | $min(|\theta|)$ | random | yes | yes |
| SNFS | uniform | $min(|\theta|)$ | momentum | yes | no |
| RigL | *Erdős-Rényi-kernel* | $min(|\theta|)$ | gradient | yes | no |
| Selfish-RNN | uniform | $min(|\theta|)$ | random | yes | yes |

makes the "winning tickets" work. Our work shows that there exists a much more efficient and robust way to find those "winning ticketts" without any special initialization. The aforementioned methods require at least the same training cost as training a dense model, sometimes even more, as a pre-trained dense model is involved. We compare our method with state-of-the-art pruning method proposed by Zhu & Gupta (2017) in Appendix I. With fewer training costs, our method is able to discover sparse networks that can achieve lower test perplexity than iterative pruning.

(2) *Learning Sparsity During Training.* There are also some works attempting to learn the sparse networks during training. Louizos et al. (2017b) and Wen et al. (2018) are examples that gradually enforce the network weights to zero via $L_0$ and $L_1$ regularization, respectively. Dai et al. (2018) proposed a singular value decomposition (SVD) based method to accelerate the training process for LSTMs. Liu et al. (2020a) proposed Dynamic Sparse Training to discover sparse structure by learning binary masks associated with network weights. However, these methods start with a fully dense network, and hence are not memory efficient.

(3) *One-Shot Pruning.* Some works aim to find sparse neural networks by pruning once prior to the main training phase based on some salience criteria, such as connection sensitivity (Lee et al., 2019), signal propagation, (Lee et al., 2020), and gradient signal preservation (Wang et al., 2020). These techniques can find sparse networks before the standard training, but at least one iteration of dense model needs to be trained to identify the sparse sub-networks, and therefore the pruning process is not applicable to memory-limited scenarios. Additionally, one-shot pruning generally cannot match the performance of dynamic sparse training, especially at extreme sparsity levels (Wang et al., 2020).

**Sparse-to-Sparse.** Recently, many works have emerged to train intrinsically sparse neural networks from scratch to obtain efficiency both for training and inference.

(1) *Static Sparse Training.* Mocanu et al. (2016) introduced intrinsically sparse networks by exploring the scale-free and small-world topological properties in Restricted Boltzmann Machines. Later, some works expand static sparse training into CNNs based on expander graphs and show comparable performance (Prabhu et al., 2018; Kepner & Robinett, 2019).

(2) *Dynamic Sparse Training.* Mocanu et al. (2018) introduced Sparse Evolutionary Training (SET) which initializes a sparse network and dynamically changes the sparse connectivity by a simple remove-and-regrow strategy. At the same time, DeepR (Bellec et al., 2018) trained very sparse networks by sampling the sparse connectivity based on a Bayesian posterior. The iterative configuration updates have been proved to converge to a stationary distribution. Mostafa & Wang (2019) introduced Dynamic Sparse Reparameterization (DSR) to train sparse neural networks while dynamically adjusting the sparsity levels of different layers. Sparse Networks from Scratch (SNFS) (Dettmers & Zettlemoyer, 2019) improved the sparse training performance by growing free weights according to their momentum. It requires extra computation and memory to update the dense momentum tensor for each iteration. Further, Evci et al. (2020) introduced RigL which activates weights with the highest magnitude gradients. This approach grows weights expected to receive gradients with high magnitudes, while amortizing a large number of memory requirements and computational cost caused by momentum. Due to the inherent limitations of deep learning software and hardware libraries, all of the above works simulate sparsity using a binary mask over weights. More recently, Liu et al. (2020b) proved the potentials of DST by developing for the first time an independent software framework to train very large truly sparse MLPs trained with SET. However, all these works mainly focus on CNNs and MLPs, and they are not designed to match state-of-the-art performance for RNNs.

We summarize the properties of all approaches compared in this paper in Table 1. Same with SET, our method can guarantee *Backward Sparse*, which does not require any extra information from the removed weights. Additionally, we discuss the differences among SET, pruning techniques, and our method in Appendix H.

## 3 SPARSE RNN TRAINING

Our sparse RNN training method is illustrated in Figure 1 with LSTM as a specific case of RNNs. Note that our method can be easily applied to any other RNN variants. The only difference is the number of cell weights. Before training, we randomly initialize each layer at the same sparsity (the fraction of zero-valued weights), so that the training costs are proportional to the dense model at the beginning. To explore more sparse structures, while to maintain a fixed sparsity level, we need to optimize the sparse connectivity together with the corresponding weights (a combinatorial optimization problem). We apply dynamic sparse connectivity and SNT-ASGD to handle this combinatorial optimization problem. The pseudocode of the full training procedure of our algorithm is shown in Algorithm 1.

### 3.1 DYNAMIC SPARSE CONNECTIVITY

We consider uniform sparse initialization, magnitude weight removal, random weight growth, cell weight redistribution together as main components of our dynamic sparse connectivity method, which can ensure a fixed number of parameters and a clear sparse backward pass, as discussed next.

**Notation.** Given a dataset of $N$ samples $\mathbf{D} = \{(x_i, y_i)\}_{i=1}^N$ and a network $f(x; \theta)$ parameterized by $\theta$. We train the network to minimize the loss function $\sum_{i=1}^N L(f(x_i; \theta), y_i)$. The basic mechanism of sparse neural networks is to use a fraction of parameters to reparameterize the whole network, while preserving the performance as much as possible. Hence, a sparse neural network can be denoted as $f_s(x; \theta_s)$ with a sparsity level $S = 1 - \frac{\|\theta_s\|_0}{\|\theta\|_0}$, where $\|\cdot\|_0$ is the $\ell_0$-norm.

**Uniform Sparse Initialization.** First, the network is uniformly initialized with a sparse distribution in which the sparsity level of each layer is the same S. More precisely, the network is initialized by:

$$\theta_s = \theta \odot M \tag{1}$$

where $\theta$ is a dense weight tensor initialized in a standard way; $M$ is a binary tensor, in which nonzero elements are sampled uniformly based on the sparsity $S$; $\odot$ refers to the Hadamard product.

**Magnitude Weight Removal.** For non-RNN layers, we use magnitude weight removal followed by random weight growth to update the sparse connectivity. We remove a fraction $p$ of weights with the smallest magnitude after each training epoch. This step is performed by changing the binary tensor $M$, as follows:

$$M = M - P \tag{2}$$

where $P$ is a binary tensor with the same shape as $M$, in which the nonzero elements have the same indices with the top-$p$ smallest-magnitude nonzero weights in $\theta_s$, with $\|P\|_0 = p\|M\|_0$.

**Random Weight Growth.** To keep a fixed parameter count, we randomly grow the same number of weights immediately after weight removal, by:

$$M = M + R \tag{3}$$

where $R$ is a binary tensor where the nonzero elements are randomly located at the position of zero elements of $M$. We choose random growth to get rid of using any information of the non-existing weights, so that both feedforward and backpropagation are completely sparse. It is more desirable to have such pure sparse structures as it enables the possibility of conceiving in the future specialized hardware accelerators for sparse neural networks. Besides, our analysis of growth methods in Section 4.3 shows that random growth can explore more sparse structural degrees of freedom than gradient growth, which might be crucial to the sparse training.

**Cell Weight Redistribution.** Our dynamic sparse connectivity differs from previous methods mainly in cell weight redistribution. For RNN layers, the naive approach is to sparsify all cell weight tensors independently at the same sparsity, as shown in Liu et al. (2019) which is a straightforward extension of applying SET to RNNs. Essentially, it is more desirable to redistribute new parameters to cell weight tensors dependently, as all cell weight tensors collaborate together to regulate information. Intuitively, we redistribute new parameters in a way that weight tensors containing more large-magnitude weights should have more parameters. Large-magnitude weights indicate that their loss

gradients are large and few oscillations occur. Thus, weight tensors with more large-magnitude connections should be reallocated with more parameters to accelerate training. Concretely, for each RNN layer $l$, we remove weights dependently given by an ascending sort:

$$Sort_p(|\theta_1^l|, |\theta_2^l|, .., |\theta_t^l|) \qquad (4)$$

where $\{\theta_1^l, \theta_2^l, ..., \theta_t^l\}$ are all weight tensors within each cell, and $Sort_p$ returns $p$ indices of the smallest-magnitude weights. After weight removal, new parameters are uniformly grown to each weight tensor to implement our cell weight redistribution gradually. We also tried other approaches including the mean value of the magnitude of nonzero weights or the mean value of the gradient magnitude of nonzero weights, but our approach achieves the best performance, as shown in Appendix B. We further demonstrate the final sparsity breakdown of cell weights learned by our method in Appendix M and observe that weights of forget gates are consistently sparser than other weights for all models. Note that redistributing parameters across cell weight tensors does not change the FLOP counting, as the sparsity of each layer is not changed. In contrast, the across-layer weight redistribution used by DSR and SNFS affects the sparsity level of each layer. As a result, it will change the number of floating-point operations (FLOPs).

Similar with SNFS, We also decay the removing rate $p$ to zero with a cosine annealing. We further use Eq. (1) to enforce the sparse structure before the forward pass and after the backward pass, so that the zero-valued weights will not contribute to the loss. And all the newly activated weights are initialized to zero.

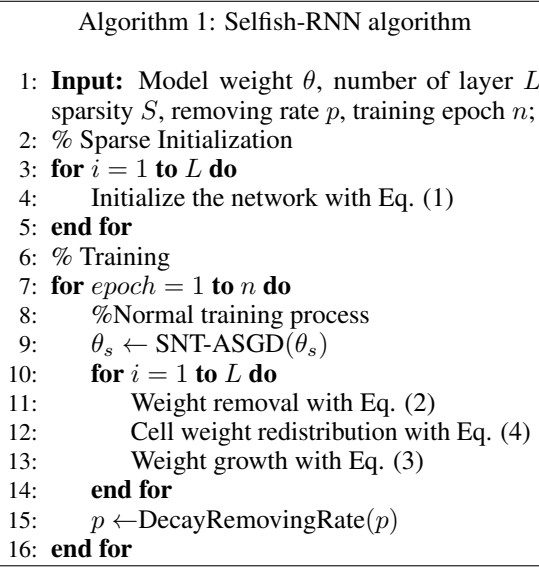

Algorithm 1: Selfish-RNN algorithm

1: **Input:** Model weight $\theta$, number of layer $L$, sparsity $S$, removing rate $p$, training epoch $n$;
2: % Sparse Initialization
3: **for** $i = 1$ **to** $L$ **do**
4:     Initialize the network with Eq. (1)
5: **end for**
6: % Training
7: **for** $epoch = 1$ **to** $n$ **do**
8:     %Normal training process
9:     $\theta_s \leftarrow \text{SNT-ASGD}(\theta_s)$
10:     **for** $i = 1$ **to** $L$ **do**
11:         Weight removal with Eq. (2)
12:         Cell weight redistribution with Eq. (4)
13:         Weight growth with Eq. (3)
14:     **end for**
15:     $p \leftarrow \text{DecayRemovingRate}(p)$
16: **end for**

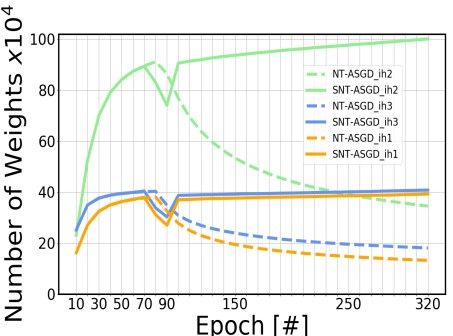

Figure 2: The number of weights whose magnitude is larger than 0.1 during training for ON-LSTM. The solid lines represent SNT-ASGD and dashed lines represent standard NT-ASGD. *ih1*, *ih2*, and *ih3* refer to input weights in the first, second and third LSTM layer.

## 3.2 SPARSE NON-MONOTONICALLY TRIGGERED ASGD

Non-monotonically Triggered ASGD (NT-ASGD) has been shown to achieve surprising performance with various RNNs (Merity et al., 2018; Yang et al., 2018; Shen et al., 2019). However, it becomes less appealing for sparse RNNs training. Unlike dense networks in which every parameter in the model is updated at each iteration, for sparse networks, the zero-valued weights remain zero when they are not activated. Once these zero-valued weights are activated, the original averaging operation of standard NT-ASGD will immediately bring them close to zero. Thereby, after the averaging operation is triggered, the number of valid weights will decrease sharply as shown in Figure 2. To alleviate this problem, we introduce SNT-ASGD as following:

$$\tilde{w}_i = \begin{cases} 0 & if \ m_i = 0, \forall i, \\ \frac{\sum_{t=T_i}^{K} w_{i,t}}{(K - T_i + 1)} & if \ m_i = 1, \forall i. \end{cases} \qquad (5)$$

where $\tilde{w}_i$ is the value returned by SNT-ASGD for weight $w_i$; $w_{i,t}$ represents the actual value of weight $w_i$ at the $t^{th}$ iteration; $m_i = 1$ if the weight $w_i$ exists and $m_i = 0$ means that the weight $w_i$ does not exist; $T_i$ is the iteration in which the weight $w_i$ grows most recently; and $K$ is the total

number of iterations. We demonstrate the effectiveness of SNT-ASGD in Figure 2. At the beginning, trained with SGD, the number of weights with high magnitude increases fast. However, the trend starts to descend significantly once the optimization switches to NT-ASGD at the $80^{th}$ epoch, whereas the trend of SNT-ASGD continues to rise after a small drop caused by the averaging operation.

To better understand how proposed components, cell weight redistribution and SNT-ASGD, improve the sparse RNN training performance, we further conduct an ablation study in Appendix A. It is clear to see that both of them lead to significant performance improvement.

## 4 EXPERIMENTAL RESULTS

We evaluate Selfish-RNN with various models including stacked LSTMs, RHNs, ON-LSTM on the Penn TreeBank dataset and AWD-LSTM-MoS on the WikiText-2 dataset. The performance of Selfish-RNN is compared with 5 state-of-the-art sparse inducing techniques, including Intrinsic Sparse Structures (ISS) (Wen et al., 2018), SET, DSR, SNFS, and RigL. ISS is a method to explore sparsity inside RNNs by using group Lasso regularization. We choose Adam (Kingma & Ba, 2014) optimizer for SET, DSR, SNFS, and RigL. We also evaluate our methods with two state-of-the-art RNN models, ON-LSTM on PTB and AWD-LSTM-MoS on Wikitext-2, as reported in Appendix D and Appendix E, respectively.

### 4.1 STACKED LSTMS

Table 2: Single model perplexity on validation and test sets for the Penn Treebank language modeling task with Stacked LSTMs and RHNs. FLOPs required to train the entire model and to test on single sample are reported. Methods with superscript '*' indicates reported results in the corresponding papers. "Static-ER" and "Static-uni" are the static sparse network trained from scratch with ER distribution and uniform distribution, respectively. "Small" refers the small-dense network.

| | Stacked LSTMs | | | | RHN | | | |
|---|---|---|---|---|---|---|---|---|
| Models | FLOPs (Train) | FLOPs (Test) | Val | Test | FLOPs (Train) | FLOPs (Test) | Val | Test |
| Dense* | 1x(3.1e16) | 1x(7.2e10) | 82.57 | 78.57 | 1x(6.5e16) | 1x(3.3e10) | 67.90 | 65.40 |
| Dense (NT-ASGD) | 1x | 1x | 74.51 | 72.40 | 1x | 1x | 63.44 | 61.84 |
| | S=0.67 | | | | S=0.53 | | | |
| Small (NT-ASGD) | 0.33x | 0.33x | 88.67 | 86.33 | 0.47x | 0.47x | 70.10 | 68.40 |
| Static-ER (SNT-ASGD) | 0.33x | 0.34x | 81.02 | 79.30 | 0.47x | 0.47x | 75.74 | 73.21 |
| Static-uni (SNT-ASGD) | 0.33x | 0.33x | 80.37 | 78.61 | 0.47x | 0.47x | 74.11 | 71.83 |
| ISS | 0.28x | 0.20x | 80.70 | 76.78 | 0.50x | 0.47x | 68.10 | 65.40 |
| SET | 0.33x | 0.34x | 87.30 | 85.49 | 0.47x | 0.47x | 63.66 | 61.08 |
| DSR | 0.38x | 0.40x | 89.95 | 88.16 | 0.47x | 0.47x | 65.38 | 63.19 |
| SNFS | 0.63x | 0.38x | 88.31 | 86.28 | 0.63x | 0.45x | 74.02 | 70.99 |
| RigL | 0.33x | 0.34x | 88.39 | 85.61 | 0.47x | 0.47x | 67.43 | 64.41 |
| RigL (SNT-ASGD) | 0.33x | 0.34x | 78.31 | 75.90 | 0.47x | 0.47x | 64.82 | 62.47 |
| Selfish-RNN | 0.33x | 0.33x | **73.76** | **71.65** | 0.47x | 0.47x | **62.10** | **60.35** |
| | S=0.62 | | | | S=0.68 | | | |
| ISS | 0.32x | 0.23x | 78.67 | 75.53 | 0.34x | 0.32x | 70.30 | 67.70 |
| RigL (SNT-ASGD) | 0.38x | 0.39x | 77.16 | 74.76 | 0.32x | 0.32x | 69.32 | 66.64 |
| Selfish-RNN | 0.38x | 0.38x | **73.50** | **71.42** | 0.32x | 0.32x | **66.35** | **64.03** |

As introduced by Zaremba et al. (2014), stacked LSTMs (large) is a two-layer LSTM model with 1500 hidden units for each LSTM layer. We choose the same sparsity as ISS, 67% and 62%. We empirically found that 0.7 is a safe choice for the removing rate of stacked LSTMs. The clip norm is set to 0.25 and all models are trained for 100 epochs.

Results are shown in the left side of Table 2. To evaluate our sparse training method fairly, we also provide a new dense baseline trained with the standard NT-ASGD, achieving 6 lower test perplexity than the widely-used baseline. We also test whether a small dense network and a static sparse network

with the same number of parameters as Selfish-RNN can match the performance of Selfish-RNN. We train a dense stacked LSTMs with 700 hidden units, named as "Small". In line with the previous studies (Mocanu et al., 2018; Mostafa & Wang, 2019; Evci et al., 2020), both static sparse networks and the small-dense network fail to match the performance of Selfish-RNN. Training a static sparse network from scratch with *uniform* distribution performs better than the one with *ER* distribution. Trained with Adam, all sparse training techniques fail to match the performance of ISS and dense models. Models trained with SNT-ASGD obtain substantially lower perplexity, and Selfish-RNN achieves the lowest one, even better than the new dense baseline with much fewer training costs.

Table 3: Performance comparison of all DST methods trained with Adam, momentum SGD and SNT-ASGD with stacked LSTMs on PTB.

| Models | #Param | FLOPs (Train) | FLOPs (Test) | Val | Test |
|---|---|---|---|---|---|
| Dense* | 66.0M | 1x (3.1e16) | 1x (7.2e10) | 82.57 | 78.57 |
| Dense (NT-ASGD) | 66.0M | 1x | 1x | 74.51 | 72.40 |
| | | S=0.67 | | | |
| DSR (Adam) | 21.8M | 0.38x | 0.40x | 89.95 | 88.16 |
| SNFS (Adam) | 21.8M | 0.63x | 0.38x | 88.31 | 86.28 |
| RigL (Adam) | 21.8M | 0.33x | 0.34x | 88.39 | 85.61 |
| SET (Adam) | 21.8M | 0.33x | 0.34x | 87.30 | 85.49 |
| Selfish-RNN (Adam) | 21.8M | 0.33x | 0.33x | **85.70** | **82.85** |
| SNFS (Momentum SGD) | 21.8M | 0.61x | 0.36x | 90.09 | 87.98 |
| SET (Momentum SGD) | 21.8M | 0.33x | 0.34x | 85.73 | 82.52 |
| RigL (Momentum SGD) | 21.8M | 0.33x | 0.34x | 84.78 | 80.81 |
| DSR (Momentum SGD) | 21.8M | 0.32x | 0.32x | 82.89 | 80.09 |
| Selfish-RNN (Momentum SGD) | 21.8M | 0.33x | 0.33x | **82.48** | **79.69** |
| SNFS (SNT-ASGD) | 21.8M | 0.63x | 0.40x | 82.11 | 79.50 |
| RigL (SNT-ASGD) | 21.8M | 0.33x | 0.34x | 78.31 | 75.90 |
| SET (SNT-ASGD) | 21.8M | 0.33x | 0.34x | 76.78 | 74.84 |
| Selfish-RNN (SNT-ASGD) | 21.8M | 0.33x | 0.33x | 73.76 | 71.65 |
| DSR (SNT-ASGD) | 21.8M | 0.32x | 0.32x | **72.30** | **70.76** |

To understand better the effect of different optimizers on different DST methods, we report the performance of all DST methods trained with Adam, momentum SGD, and SNT-ASGD. The learning rate of Adam is set as 0.001. The learning rate of momentum SGD is 2 decreased by a factor of 1.33 once the loss fails to decrease and the momentum coefficient is 0.9. The weight decay is set as 1.2e-6 for all optimizers. For SNFS (SNT-ASGD), we replace momentum of weights with their gradients, as SNT-ASGD does not involve any momentum terms. We use the same hyperparameters for all DST methods. The results are shown in Table 3. It is clear that SNT-ASGD brings significant perplexity improvements to all sparse training techniques. This further stands as empirical evidence that SNT-ASGD is crucial to improve the sparse training performance in the RNN setting. Moreover, compared with other DST methods, Selfish-RNN is quite robust to the choice of optimizers due to its simple scheme to update sparse connectivity. Advanced strategies such as across-layer weight redistribution used in DSR and SNFS, gradient-based weight growth used in RigL and SNFS heavily depend on optimizers. They might work decently for some optimization methods but may not work for others.

Additionally, note that different DST methods use different sparse distributions, leading to very different computational costs even with the same sparsity. We also report the approximated training and inference FLOPs for all methods. The FLOP gap between Selfish-RNN and RigL is very small, whereas SNFS requires more FLOPs than our method for both training and inference (see Appendix L for details on how FLOPs are calculated). ISS achieves a lower number of FLOPs, since it does not sparsify the embedding layer and therefore, their LSTM layers are much more sparse than LSTM layers obtained by other methods. This would cause a fewer number of FLOPs as LSTM layers typically require more FLOPs than other layers.

## 4.2 Recurrent Highway Networks

Recurrent Highway Networks (Zilly et al., 2017) is a variant of RNNs allowing RNNs to explore deeper architectures inside the recurrent transition. See Appendix C for experimental settings of RHN. The results are shown in the right side of Table 2. Selfish-RNN achieves better performance than the dense model with half FLOPs. Unlike the large FLOP discrepancy of stacked LSTMs, the FLOP gap between different sparse training techniques for RHNs is very small, except SNFS which requires computing dense momentum for each iteration. Additionally, ISS has similar FLOPs with Selfish-RNN for RHN, as it sparsifies the embedding layer as well.

## 4.3 Analyzing the Performance of Selfish-RNN

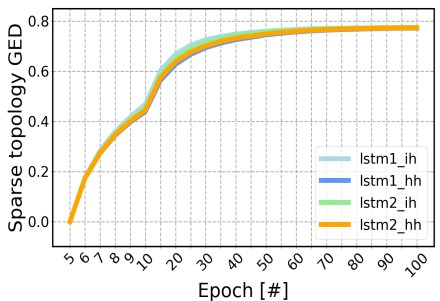 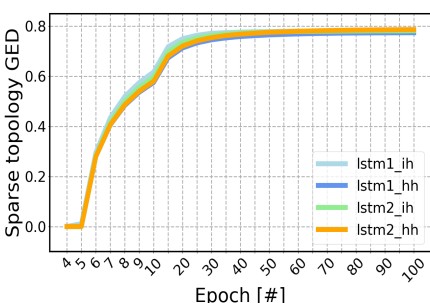

Figure 3: (**left**) One random-initialized sparse network trained with Selfish-RNN end up with a very different sparse connectivity topology. (**right**) Two same-initialized networks trained with different random seeds end up with very different sparse connectivity topologies. $ih$ is the input weight tensor comprising four cell weights and $hh$ is the hidden state weight tensor comprising four cell weights.

**Analysis of Evolutionary Trajectory of Sparse Connectivity.** The fact that Selfish-RNN consistently achieves good performance with different runs naturally raises some questions: e.g., are final sparse connectivities obtained by different runs similar or very different? Is the distance between the original sparse connectivity and the final sparse connectivity large or small? To answer these questions, we investigate a method based on graph edit distance (GED) (Sanfeliu & Fu, 1983) to measure the topological distance between different sparse connectivities learned by different runs. The distance is scaled between 0 and 1. The smaller the distance is, the more similar the two sparse topologies are (See Appendix J for details on how we measure the sparse topological distance).

The results are demonstrated in Figure 3. Figure 3-left shows how the topology of one random-initialized network evolves when trained with Selfish-RNN. We compare the topological distance between the sparse connectivity obtained at the $5^{th}$ epoch and the sparse connectivities obtained in the following epochs. We can see that the distance gradually increases from 0 to a very high value 0.8, meaning that Selfish-RNN optimizes the initial topology to a very different one after training. Moreover, Figure 3-right illustrates that the topological distance between two same-initialized networks trained with different seeds after the $4^{th}$ epoch. We can see that starting from the same sparse topology, they evolve to completely different sparse connectivities. Note that even when leading to completely different sparse connectivities, different runs achieve similarly good performance, which indicates that in the case of RNNs there exist many good local optima in terms of sparse connectivity that can have equally good performance. This phenomenon complements the findings of Liu et al. (2020c) which show that there are numerous sparse sub-networks performing similarly well in the context of MLPs.

**Analysis of Sparse Initialization.** We compare two types of sparse initialization, ER distribution and uniform distribution. Uniform distribution namely enforces the sparsity level of each layer to be the same as $S$. ER distribution allocates higher sparsity to larger layers than smaller ones. Note that its variant *Erdős-Rényi-kernel* proposed by Evci et al. (2020) scales back to ER for RNNs, as no kernels are involved. The results are shown as the *Static* group in Table 2. We can see that uniform distribution outperforms ER distribution consistently. Moreover, ER usually causes RNN layers to be less sparse than other layers, resulting in a small increase of FLOPs.

**Analysis of Growth Methods.** Methods that leverage gradient-based weight growth (SNFS and RigL) have shown superiority over the methods using random-based weight growth for CNNs. However, we observe a different behavior with RNNs. We set up a controlled experiment to compare these two methods with SNT-ASGD and momentum SGD. We report the results with various update intervals (the number of iterations between sparse connectivity updates) in Figure 4. Surprisingly, gradient-based growth performs worse than random-based growth in most cases. And there is an increased performance gap as the update interval increases. Our hypothesis is that random growth helps in exploring better the search space, as it naturally considers a large number of various sparse connectivities during training, which is crucial to the performance of dynamic sparse training. Differently, gradient growth drives the network topology towards some similar local optima for the sparse connectivity as it uses a greedy search strategy (highest gradient magnitude) at every topological change. However, benefits provided by high-magnitude gradients might change dynamically afterwards due to complicated interactions between weights. We empirically illustrate our hypothesis via the proposed distance measure between sparse connectivities in Appendix K.

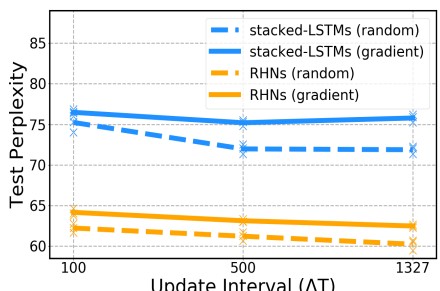 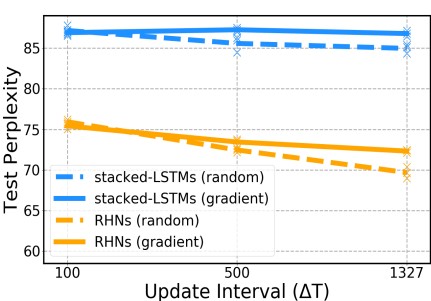

Figure 4: Comparison between random-based growth and gradient-based growth. (**left**) Models trained with SNT-ASGD. (**right**) Models trained with momentum SGD.

**Analysis of Hyper-parameters.** The sparsity $S$ and the initial removing rate $p$ are two hyper-parameters of our method. We show their sensitivity analysis in Appendix F and Appendix G. We find that Selfish Stacked LSTMs, RHNs, ON-LSTM, and AWD-LSTM-MoS need around 25%, 40%, 45%, and 40% parameters to reach the performance of their dense counterparts, respectively. And our method is quite robust to the choice of the initial removing rate.

## 5 CONCLUSION

In this paper, we proposed an approach to train sparse RNNs from scratch with a fixed parameter count throughout training. Further, we introduced SNT-ASGD, a specially designed sparse optimizer for training sparse RNNs and we showed that it substantially improves the performance of all dynamic sparse training methods in RNNs. We observed that random-based growth achieves lower perplexity than gradient-based growth in the case of RNNs. Further, we developed an approach to compare two different sparse connectivities from the perspective of graph theory. Using this approach, we found that random-based growth explores better the topological search space for optimal sparse connectivities, whereas gradient-based growth is prone to drive the network towards similar sparse connectivity patterns. opening the path for a better understanding of sparse training.

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

## A  ABLATION STUDY

To verify if the improvement shown above is caused by the cell weight redistribution or the Sparse NT-ASGD, we conduct an ablation study for all architectures. To avoid distractive factors, all models use the same hyper-parameters with the ones reported in the paper. And the use of finetuning is not excluded. We present the validation and testing perplexity for variants of our model without these two contributions, as shown in Table 4. Not surprisingly, removing either of these two novelties degrades the performance. There is a significant degradation in the performance for all models, up to 13 perplexity point, if the optimizer switches to the standard NT-ASGD. This stands as empirical evidence regarding the benefit of SNT-ASGD. Without cell weight redistribution, the testing perplexity also rises. The only exception is RHN whose number of redistributed weights in each layer is only two. This empirically shows that cell weight redistribution is more effective for the models with more cell weights.

Table 4: Model ablation of test perplexity for Selfish-RNN for stacked LSTMs, RHNs, ON-LSTM on Penn Treebank and AWD-LSTM-MoS on WikiText-2.

| Methods | Stacked LSTMs | RHNs | ON-LSTM | AWD-LSTM-MoS |
|---|---|---|---|---|
| Selfish-RNN | 71.65 | 60.35 | 55.68 | 63.05 |
| w/o Cell weight redistribution | 72.89 | 60.26 | 57.48 | 65.27 |
| w/o Sparse NT-ASGD | 73.74 | 69.70 | 69.28 | 71.65 |

## B  COMPARISON OF DIFFERENT CELL WEIGHT REDISTRIBUTION METHODS

In Table 5, we conduct a small experiment to compare different methods of cell weight redistribution with stacked LSTMs, including redistributing based on the mean value of the magnitude of nonzero weights from different cell weights and the mean value of the gradient magnitude of nonzero weights.

Table 5: A small experiment about the comparison among different cell weight redistribution methods. The experiment is evaluated with stacked LSTMs on Penn Treebank.

| Cell weight redistribution | #Param | Validation | Test |
|---|---|---|---|
| Mean of the magnitude of nonzero weights | 21.8M | 74.04 | 72.40 |
| Mean of the gradient magnitude of nonzero weights | 21.8M | 74.54 | 72.31 |
| Ours | 21.8M | **73.76** | **71.65** |

## C  EXPERIMENTAL DETAILS FOR RHN

Recurrent Highway Networks (Zilly et al., 2017) is a variant of RNNs allowing RNNs to explore deeper architecture inside the recurrent transition. Instead of stacking recurrent layers directly, RHN stacks multiple highway layers on top of recurrent state transition. Within each highway layer, free weights are redistributed across the input weight and the state weight. The sparsity level is set the same as ISS, 67.7% and 52.8%. Dropout rates are set to be 0.20 for the embedding layer, 0.65 for the input, 0.25 for the hidden units, and 0.65 for the output layer. The model is trained for 500 epochs with a learning rate of 15, a batch size of 20, and a sequence length to of 35. At the end of each training epoch, new weights are redistributed across the weights of the $H$ nonlinear transform and the $T$ gate.

## D  EXPERIMENTAL RESULTS WITH ON-LSTM

Table 6: Single model perplexity on validation and test sets for the Penn Treebank language modeling task with ON-LSTM. Methods with "ASGD" are trained with SNT-ASGD. The numbers reported are averaged over five runs.

| Models | #Param | Val | Test |
|---|---|---|---|
| $\text{Dense}_{1000}$ | 25M | $58.29 \pm 0.10$ | $56.17 \pm 0.12$ |
| $\text{Dense}_{1300}$ | 25M | $58.55 \pm 0.11$ | $56.28 \pm 0.19$ |
| SET | 11.3M | $65.90 \pm 0.08$ | $63.56 \pm 0.14$ |
| DSR | 11.3M | $65.22 \pm 0.07$ | $62.55 \pm 0.06$ |
| SNFS | 11.3M | $68.00 \pm 0.10$ | $65.52 \pm 0.15$ |
| RigL | 11.3M | $64.41 \pm 0.05$ | $62.01 \pm 0.13$ |
| $\text{RigL}_{1000}$ (ASGD) | 11.3M | $59.17 \pm 0.08$ | $57.23 \pm 0.09$ |
| $\text{RigL}_{1300}$ (ASGD) | 11.3M | $59.10 \pm 0.05$ | $57.44 \pm 0.15$ |
| $\text{Selfish-RNN}_{1000}$ | 11.3M | $58.17 \pm 0.06$ | $56.31 \pm 0.10$ |
| $\text{Selfish-RNN}_{1300}$ | 11.3M | $\mathbf{57.67 \pm 0.03}$ | $\mathbf{55.82 \pm 0.11}$ |

Table 7: Single model perplexity on validation and test sets for the WikiText-2 language modeling task with AWD-LSTM-MoS. Baseline is AWD-LSTM-MoS obtained from Yang et al. (2018). Methods with "ASGD" are trained with SNT-ASGD.

| Models | #Param | Val | Test |
|---|---|---|---|
| Dense | 35M | 66.01 | 63.33 |
| SET | 15.6M | 72.82 | 69.61 |
| DSR | 15.6M | 69.95 | 66.93 |
| SNFS | 15.6M | 79.97 | 76.18 |
| RigL | 15.6M | 71.36 | 68.52 |
| RigL (ASGD) | 15.6M | 68.84 | 65.18 |
| Selfish-RNN | 15.6M | $\mathbf{65.96}$ | $\mathbf{63.05}$ |

Proposed by Shen et al. (2019) recently, ON-LSTM can learn the latent tree structure of natural language by learning the order of neurons. For a fair comparison, we use exactly the same model hyper-parameters and regularization used in ON-LSTM. We set the sparsity of each layer to 55% and the initial removing rate to 0.5. We train the model for 1000 epochs and rerun SNT-ASGD as a fine-tuning step once at the $500^{th}$ epoch, dubbed as $\text{Selfish-RNN}_{1000}$. As shown in Table 6, Selfish-RNN outperforms the dense model while reducing the model size to 11.3M. Without SNT-ASGD, sparse training techniques can not reduce the test perplexity to 60. SNT-ASGD is able to improve the performance of RigL by 5 perplexity. Moreover, one interesting observation is that one of the regularizations used in the standard ON-LSTM, DropConnect, is perfectly compatible with our method, although it also drops the hidden-to-hidden weights out randomly during training.

In our experiments we observe that Selfish-RNN benefits significantly from the second fine-tuning operation. We scale the learning schedule to 1300 epochs with two fine-tuning operations after 500 and 1000 epochs, respectively, dubbed as $\text{Selfish-RNN}_{1300}$. It is interesting that $\text{Selfish-RNN}_{1300}$ can achieve lower testing perplexity after the second fine-tuning step, whereas the dense model $\text{Dense}_{1300}$ can not even reach again the perplexity that it had before the second fine-tuning. The heuristic explanation here is that our method helps the optimization escape the local optima or a local saddle point by optimizing the sparse structure, while for dense models whose energy landscape is fixed, it is very difficult for the optimizer to find its way off the saddle point or the local optima.

## E  EXPERIMENTAL RESULTS WITH AWD-LSTM-MoS

We also evaluate Selfish-RNN on the WikiText-2 dataset. The model we choose is AWD-LSTM-MoS (Yang et al., 2018), which is the state-of-the-art RNN-based language model. It replaces Softmax with *Mixture of Softmaxes* (MoS) to alleviate the Softmax bottleneck issue in modeling natural language. For a fair comparison, we exactly follow the model hyper-parameters and regularization used in AWD-LSTM-MoS. We sparsify all layers with 55% sparsity except for the prior layer as its number of parameters is negligible. We train our model for 1000 epochs without finetuning or dynamical evaluation (Krause et al., 2018) to simply show the effectiveness of our method. As demonstrated in Table 7. Selfish AWD-LSTM-MoS can reach dense performance with 15.6M parameters.

## F  EFFECT OF SPARSITY

There is a trade-off between the sparsity level S and the test perplexity of Selfish-RNN. When there are too few parameters, the sparse neural network will not have enough capacity to model the data. If the sparsity level is too small, the training acceleration will be small. Here, we analyze this trade-off by varying the sparsity level while keeping the other experimental setup the same, as shown in

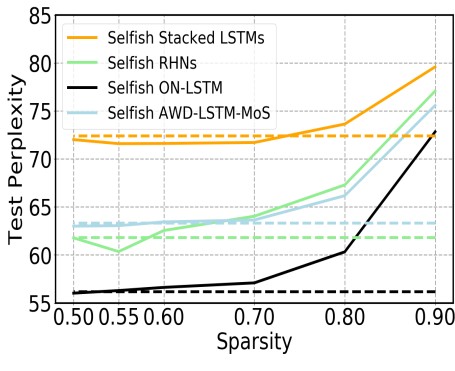 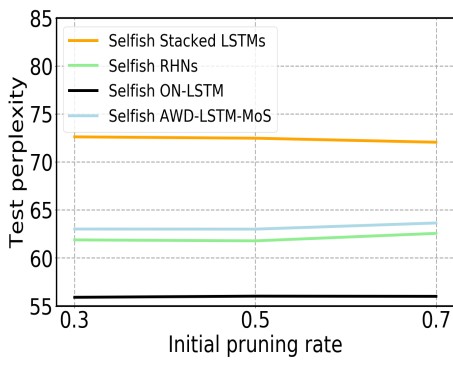

(a) Performance Sensitivity of Sparsity      (b) Performance Sensitivity of Removing rate

Figure 5: Sensitivity analysis for sparsity levels and the initial removing rate for Selfish stacked LSTMs, RHNs, ON-LSTM, and AWD-LSTM-MoS. (a) Test perplexity of all models with various sparsity levels. The initial removing rate of stacked LSTMs is 0.7, the rest is 0.5. The dashed lines represent the performance of dense models. (b) Test perplexity of all models with different initial removing rates. The sparsity level is 67%, 52.8%, 55% and 55% for Selfish stacked LSTMs, RHNs, ON-LSTM, and AWD-LSTM-MoS, respectively.

Figure 5a. We find that Selfish Stacked LSTMs, RHNs, ON-LSTM, and AWD-LSTM-MoS need around 25%, 40%, 45%, and 40% parameters to reach the performance of their dense counterparts, respectively. Generally, the performance of sparsified models is decreasing as the sparsity level increases.

## G  EFFECT OF INITIAL REMOVING RATE

The initial removing rate $p$ determines the number of removed weights at each connectivity update. We study the performance sensitivity of our algorithm to the initial removing rate $p$ by varying it $\in [0.3, 0.5, 0.7]$. We set the sparsity level of each model as the one having the best performance in Figure 5a. Results are shown in Figure 5b. We can clearly see that our method is very robust to the choice of the initial removing rate.

## H  DIFFERENCE AMONG SET, SELFISH-RNN AND ITERATIVE PRUNING METHODS

The topology update strategy of Selfish-RNN differs from SET in several important features. (1) we automatically redistribute weights across cell weights for better regularization, (2) we use magnitude-based removal instead of removing a fraction of the smallest positive weights and the largest negative weights, (3) we use uniform initialization rather than non-uniform sparse distribution like ER or ERK, as it consistently achieves better performance. Additionally, the optimizer proposed in this work, SNT-ASGD, brings substantial perplexity improvement to the sparse RNN training.

Figure 6-left illustrates a high-level overview from an efficiency perspective of the difference between Selfish-RNN and iterative pruning techniques (Han et al., 2016; Zhu & Gupta, 2017; Frankle & Carbin, 2019). The conventional pruning and re-training techniques usually involve three steps: (1) pre-training a dense model, (2) pruning unimportant weights, and (3) re-training the pruned model to improve performance. The pruning and re-training cycles can be iterated. This iteration is taking place at least once, but it may also take place several times depending on the specific algorithms used. Therefore, the sparse networks obtained via iterative pruning at least involve pre-training a dense model. Different from the aforementioned three-step techniques, FLOPs required by Selfish-RNN is proportional to the density of the model, as it allows us to train a sparse network with a fixed number of parameters throughout training in one single run, without any re-training phases. Moreover, the overhead caused by the adaptive sparse connectivity operation is negligible, as it is operated only once per epoch.

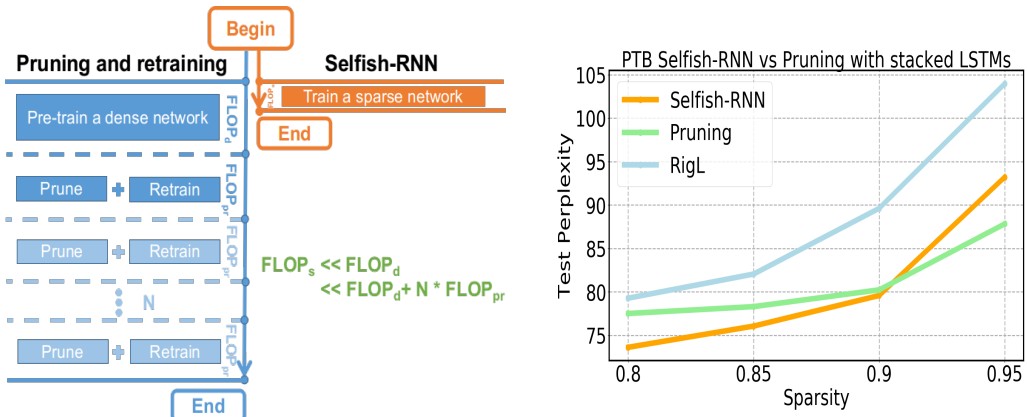

Figure 6: (**left**) A high-level overview of the difference between Selfish-RNN and the iterative pruning and re-training techniques. Blocks with light blue color represent optional pruning and retraining steps chosen based on specific approaches. (**right**) Comparison of the single model perplexity on test sets of Selfish-RNN, RigL and iterative magnitude pruning with stacked LSTMs on PTB.

## I    COMPARISON BETWEEN SELFISH-RNN AND PRUNING

It has been shown by Evci et al. (2020) that while state-of-the-art sparse training method (RigL) achieves promising performance in terms of CNNs, it fails to match the performance of pruning in RNNs. Given the fact that magnitude pruning has become a widely-used and strong baseline for model compression, we also report a comparison between Selfish-RNN and iterative magnitude pruning with stacked LSTMs. The pruning baseline here is the Tensorflow Model Pruning library (Zhu & Gupta, 2017). The results are demonstrated in Figure 6-right.

We can see that Selfish-RNN exceeds the performance of pruning in most cases. An interesting phenomenon is that, with increased sparsity, we see a decreased performance gap between Selfish-RNN and pruning. Especially, Selfish-RNN performs worse than pruning when the sparsity level is 95%. This can be attributed to the poor trainability problem of sparse models with extreme sparsity levels. Noted in Lee et al. (2020), the extreme sparse structure can break dynamical isometry (Saxe et al., 2014) of sparse networks, which degrades the trainability of sparse neural networks. Different from sparse training methods, pruning operates from a dense network and thus, does not have this problem.

## J    SPARSE TOPOLOGY DISTANCE MEASUREMENT

Our sparse topology distance measurement considers the unit alignment based on a *semi-matching* technique introduced by Li et al. (2016) and a graph distance measurement based on graph edit distance (GED) (Sanfeliu & Fu, 1983). More specifically, our measurement includes the following steps:

Step 1: We train two sparse networks with dynamic sparse training on the training dataset and store the sparse topology after each epoch. Let $W_l^i$ be the set of sparse topologies for the $l$-th layer of network $i$.

Step 2: Using the saved model, we compute the activity output on the test data, $O_l^i \in \mathbb{R}^{n \times m}$, where $n$ is the number of hidden units and $m$ is the number of samples.

Step 3: We leverage the activity units of each layer to pair-wisely match topologies $W_l^i$. We achieve unit matching between a pair of networks by finding the unit in one network with the maximum correlation to the one in the other network.

Step 4: After alignment, we apply graph edit distance (GED) to measure the similarity between pairwise $W_l^i$. Eventually, the distance is scaled to lie between 0 and 1. The smaller the distance is, the more similar the two sparse topologies are.

Here, We choose stacked LSTMs on PTB dataset as a specific case to analyze. Specifically, we train two stacked LSTMs for 100 epochs with different random seeds. We choose a relatively small removing rate of 0.1. We start alignment at the $5^{th}$ epoch to ensure a good alignment result, as at the very beginning of training networks do not learn very well. We then use the matched order of output tensors to align the pairwise topologies $W_l^i$.

## K    TOPOLOGICAL DISTANCE OF GROWTH METHODS

In this section, we empirically illustrate that gradient growth drives different networks into some similar connectivity patterns based on the proposed distance measurement between sparse connectivities. The initial removing rates are set as 0.1 for all training runs in this section. First, we measure the topological distance between two different training runs trained with gradient growth and random growth, respectively, as shown in Figure 7. We can see that, starting with very different sparse connectivity topologies, two networks trained with random growth end up at the same distance, whereas the topological distance between networks trained with gradient growth is continuously decreasing and this tendency is likely to continue as the training goes on. We further report the distance between two networks with same initialization but different training seeds when trained with gradient growth and random growth, respectively. As shown in Figure 8, the distance between sparse networks discovered by gradient growth is smaller than the distance between sparse networks discovered by random growth. These observations are in line with our hypothesis that gradient growth drives networks into some similar structures, whereas random growth explores more sparse structures spanned over the dense networks.

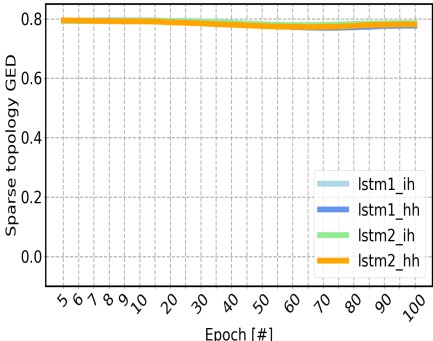 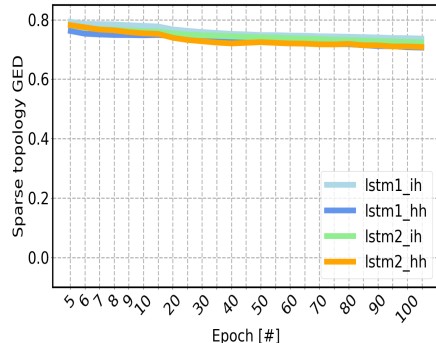

Figure 7: (**left**) The topological distance between two different training runs of stacked LSTMs trained with random growth. (**right**) The topological distance between two different training runs of stacked LSTMs trained with gradient growth.

## L    FLOPS ANALYSIS OF DIFFERENT APPROACHES

We follow the way of calculating training FLOPs layer by layer based on sparsity level $s^l$, proposed by Evci et al. (2020). We split the process of training a sparse recurrent neural network into two steps: *forward pass* and *backward pass*.

***Forward pass***    In order to calculate the loss of the current models given a batch of input data, the output of each layer is needed to be calculated based on a linear transformation and a non-linear activation function. Within each RNN layer, different cell weights are used to regulate information in sequence using the output of the previous time step and the input of this time step.

***Backward pass***    In order to update weights, during the backward pass, each layer calculates 2 quantities: the gradient of the loss function with respect to the activations of the previous layer and the gradient of the loss function with respect to its own weights. Therefore, the computational expense of *backward pass* is twice that of *forward pass*. Given that RNN models usually contain an embedding layer from which it is very efficient to pick a word vector, for models not using weight tying, we

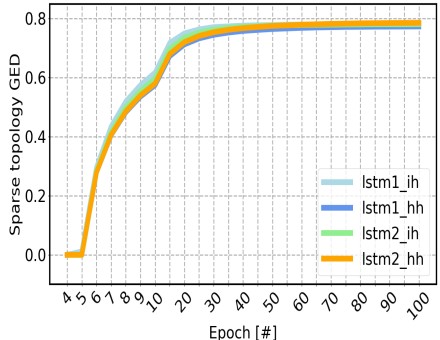 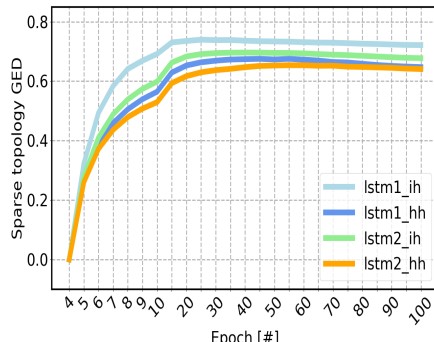

Figure 8: (**left**) The topological distance between two stacked LSTMs with same initialization but different training seeds trained with random growth. (**right**) The topological distance between two networks with same initialization but different training seeds trained with gradient growth. $ih$ is the input weight tensor comprising four cell weights and $hh$ is the hidden state weight tensor comprising four cell weights.

Table 8: Training FLOPs analysis of different sparse training approaches. $f_D$ refers to the training FLOPs for a dense model to compute one single prediction in the *forward pass* and $f_S$ refers to the training FLOPs for a sparse model. $\Delta T$ is the number of iterations used by RigL to update sparse connectivity. $s_t$ is the sparsity level of the model at iteration $t$.

| Method | Forward Pass | Backward Pass | Total |
|---|---|---|---|
| Dense | $f_D$ | $2f_D$ | $3f_D$ |
| ISS | $f_D * s_t$ | $2f_D * s_t$ | $3f_D * s_t$ |
| Pruning | $f_D * s_t$ | $2f_D * s_t$ | $3f_D * s_t$ |
| SET | $f_S$ | $2f_S$ | $3f_S$ |
| DSR | $f_S$ | $2f_S$ | $3f_S$ |
| SNFS | $f_S$ | $f_S + f_D$ | $2f_S + f_D$ |
| RigL | $f_S$ | $\frac{(2\Delta T+1)f_S+f_D}{\Delta T+1}$ | $\frac{3f_S\Delta T+2f_S+f_D}{\Delta T+1}$ |
| Selfish-RNN (ours) | $f_S$ | $2f_S$ | $3f_S$ |

only count the computations to calculate the gradient of its parameters as the training FLOPs and we omit its inference FLOPs. For models using weight tying, both the training FLOPs and the inference FLOPs are omitted.

Given a specific architecture, we denote $f_D$ as dense FLOPs required to finish one training iteration and $f_S$ as the corresponding sparse FLOPs ($f_S \approx (1 - S)f_D$), where $S$ is the sparsity level. Thus $f_S \ll f_D$ for very sparse networks. Since different sparse training methods cause different sparse distribution, their FLOPs $f_S$ are also different from each other. We omit the FLOPs used to update the sparse connectivity, as it is only performed once per epoch. Overall, the total FLOPs required for one training update on one single sample are given in Table 8. The training FLOPs of dense-to-sparse methods like, ISS and pruning, are $3f_D * s_t$, where $s_t$ is the sparsity of the model at iteration $t$. Since dense-to-sparse methods require to train a dense model for a while, their training FLOPs and memory requirement are higher than our method. For methods that allow the sparsity of each layer dynamically changing e.g., DSR and SNFS, we approximate their training FLOPs via their final distribution, as their sparse distribution converge to the final distribution in the first few epochs. ER distribution causes a bit more inference FLOPs than uniform distribution because is allocates more weights to the RNN layers than other layers. SNFS requires extra FLOPs to calculate dense gradients during the backward pass. Although RigL also uses the dense gradients to assist weight growth, it only needs to calculate dense gradients every $\Delta T$ iterations, thus its averaged FLOPs is given by $\frac{3f_S\Delta T+2f_S+f_D}{\Delta T+1}$. Here, we simply omit the extra FLOPs required by gradient-based growth, as it is negligible compared with the whole training FLOPs.

For inference, we calculate the inference FLOPs on single sample based on the final sparse distribution learned by different methods.

## M  FINAL CELL WEIGHT SPARSITY BREAKDOWN

We further study the final sparsity level across cell weights learned automatically by our method. We find a consistent observation that the weight of forget gates, either the forget gate in the standard LSTM or the master forget gate in ON-LSTM, tend to be sparser than the weight of other gates, whereas the weight of cell gates and output gates are denser than the average, as shown in Figure 9. However, there is no big difference between weights in RHN, although the $H$ nonlinear transform weight is slightly sparser than the $T$ gate weight in most RHN layers. This phenomenon is in line with the Ablation analysis where the cell weight redistribution does not provide performance improvement for RHNs. Cell weight redistribution is more important for models with more regulating weights.

## N  LIMITATION

The aforementioned training benefits have not been fully explored, as off-the-shelf software and hardware have limited support for sparse operations. The unstructured sparsity is difficult to be efficiently mapped to the existing parallel processors. The results of our paper provide motivation for new types of hardware accelerators and libraries with better support for sparse neural networks. Nevertheless, many recent works have been developed to accelerate sparse neural networks including Gray et al. (2017); Moradi et al. (2019); Ma et al. (2019); Yang & Ma (2019); Liu et al. (2020b). For instance, NVIDIA introduces the A100 GPU enabling the Fine-Grained Structured Sparsity (NVIDIA, 2020). The sparse structure is enforced by allowing two nonzero values in every four-entry vector to reduce memory storage and bandwidth by almost $2\times$. We do not claim that Selfish-RNN is the best way to obtain sparse recurrent neural networks, but simply highlights that it is an important future research direction to develop more efficient hardware and software to benefit from sparse neural networks.

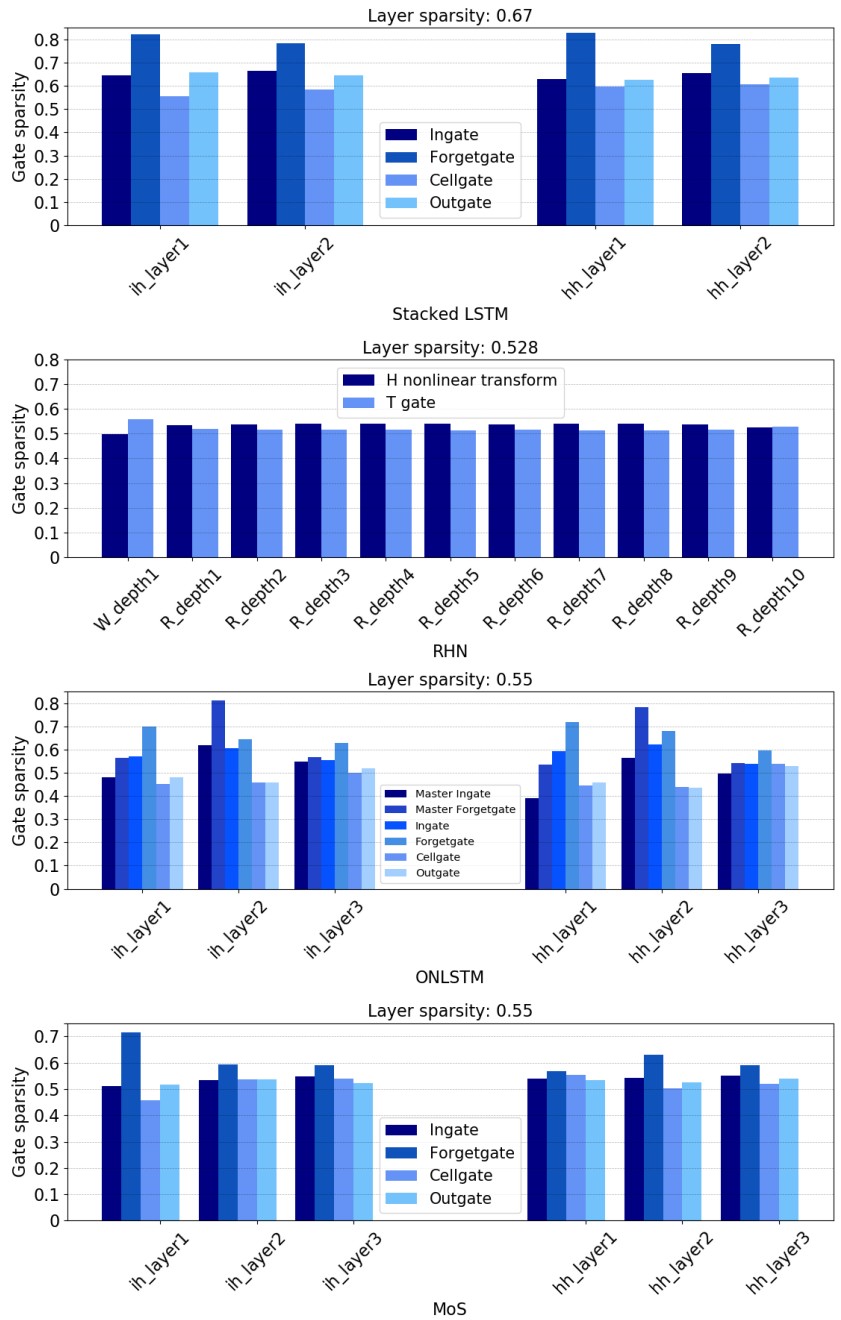

Figure 9: Breakdown of the final sparsity level of cell weights for stacked LSTMs, RHNs, ON-LSTM on PTB, AWD-LSTM-MoS on Wikitext-2. $W$ and $R$ represent the weights tensors at each layer for RHNs; $ih$ and $hh$ refer to the input weight and the hidden weight in each layer, respectively for the rest models.

