# OpenReview forum: "Selfish Sparse RNN Training"
_ICLR.cc/2021/Conference — Reject_

### Official Review · AnonReviewer1 · 2020-10-14
**lack of some important information**

**Rating:** 4
**Confidence:** 3

**Review:**

In this paper, the authors studied the possibility of sparsity exploration in Recurrent Neural Networks (RNNs) training. The main contributions include two parts: (1) Selfish-RNN training algorithm in Section 3.1 (2) SNT-ASGD optimizer in Section 3.2. The key idea of the Selfish-RNN training algorithm is a non-uniform redistribution across cell weights for better regularization. The authors mentioned previous sparse training techniques mainly focus on Multilayer Perceptron Networks (MLPs) and Convolutional Neural Networks (CNNs) rather than RNNs. This claim seems to be doubtful because one-time SVD + fine-tuning usually works very well for most RNN training applications in the industry.

Overall, this paper is carefully written and provides some interesting empirical results. However, due to the lack of some important information, it is hard to evaluate the contribution of this paper.

Here are some of my questions.

SNT-ASGD needs to save the weights w_i,t from iteration Ti to iteration K, will that cost additional memory?

The authors mentioned that they picked Adam optimizer for SET, DSR, SNFS, and RigL. Is Adam the best optimizer to build a strong baseline? I suspect Adam may not be the best optimizer for each of them.

The authors need to give more information on the hyper-parameters like the learning rate. The selection of hyper-parameters usually significantly affects the convergence/generalization performance of an RNN model. For example, the way of learning rate decay has a big impact on the performance of training Penn TreeBank dataset.

Can the authors report the training epochs and wall-clock time (e.g. in Table 2)? The sparsity typically makes modern hardware like GPUs perform poorly. That may be a concern. That’s the reason why researchers are studying structure sparsity. For future work, an analysis of computation (flops) to communication (memory access frequency) ratio seems to be necessary.

---

> ### Author Response · Authors · 2020-11-20
> **AnonReviewer1 Author Response**
>
> Thank you for your comments. We believe that there is some confusion regarding our method, which we clarify in details below:
> - **One-time SVD + fine-tuning usually works very well for most RNN training applications in the industry.** We want to re-emphasize that the central goal of our paper is to develop a method that enables training a sparse RNN from scratch (sparse-to-sparse), without involving any pre-training steps. Training a sparse neural network from scratch is a very challenging problem (please see references e.g., [1,2]), since naively training a sparse neural network with a random initialization generally leads to worse solutions than those found by iterative pruning. One-time SVD + fine-tuning starts from a dense model and the sparse model is obtained during training. It is not directly comparable to our method. Recently, various dynamic sparse training (DST) methods have been proposed to train a sparse neural network from scratch without pre-training a dense model. However, they mainly focus on MLPs and CNNs. For instance, while “The Rigged Lottery” (RigL) [3] achieves state-of-the-art sparse training results with various CNNs, it fails to match the performance of the iterative pruning in the RNN setting. In this paper, we propose Selfish-RNN to train sparse RNNs from scratch in one single run without any further fine-tuning, achieving better performance than iterative pruning.
> - **SNT-ASGD needs to save the weights $w_{i, t}$ from iteration $T_i$ to iteration $K$, will that cost additional memory?** Yes, if we would save weights $w_{i, t}$ from iteration $T_i$ to iteration $K$, that will cost a large amount of additional memory. Fortunately, instead of saving the weights at every iteration, we can simply save the total number of iterations in which a weight has been activated. Using this number, together with the previous averaged value, and the current weight, we can easily calculate the current averaged value, storing just three numbers for an weight at any-time during training.
> - **Is Adam the best optimizer to build a strong baseline? I suspect Adam may not be the best optimizer for each of them.** We agree with you that it is better to analyze different DST methods with the same optimizer. And in fact, we did provide the comparison of different DST methods trained with the same optimizer in Appendix L (We have moved this comparison in the main paper as Table 3 in the new version). It shows that our method consistently achieves very good performance, whereas the performance of other methods heavily depends on different types of optimizers.
> - **The authors need to give more information on the hyper-parameters like the learning rate.** We agree that the selection of hyper-parameters affects the convergence of RNN models. We clarify that we didn’t apply any further hyperparameter tuning techniques used in “On the State of the Art of Evaluation in Neural Language Models”. Instead, we use exactly the same hyper-parameters and regularization as the dense ON-LSTM and AWD-LSTM-MoS. Further, we extended similar settings to stacked LSTM and RHN, since there are no existing implementations of these two models trained with NT-ASGD. No need of finetuning the original hyperparameters of the dense model is another advantage of our method. For a fair comparison, the hyperparameters of all DST methods are the same.
> -**Can the authors report the training epochs and wall-clock time (e.g. in Table 2)?** The sparse structure in our paper is enforced by binary masks, as off-the-shelf software and hardware have limited support for sparse operations. However, we want to highlight the fact that our method has promising potential to reduce memory and to boost the training process. This potential has been proved until now, up to our best knowledge, just in the paper [4] in the case of multilayer perceptron. While their results are impressive, it is not straight-forward, actually it is a difficult engineering challenge, to take the same implementation approach in our case as we would need to implement all methods, optimization algorithms, activation functions, and so on, from scratch. We hope that our results will pile up on other researchers results in sparse training and soon there will be a change of perspective in such a way that the developers of deep learning software and hardware will start considering including real sparsity support in their solutions. We hope that the reviewer understands our point of view.
>
> [1] [The Difficulty of Training Sparse Neural Networks](https://arxiv.org/abs/1906.10732)
> [2] [Gradient Flow in Sparse Neural Networks and How Lottery Tickets Win](https://openreview.net/pdf?id=V1N4GEWki_E)
> [3] [Rigging the Lottery: Making All Tickets Winners](https://arxiv.org/abs/1911.11134)
> [4] [Sparse Evolutionary Deep Learning With Over One Million Artificial Neurons On Commodity Hardware](https://link.springer.com/article/10.1007/s00521-020-05136-7)

---

### Official Review · AnonReviewer3 · 2020-10-23
**Very solid paper on sparse training of RNNs which presents insights valuable for important questions on sparse training.**

**Rating:** 7
**Confidence:** 5

**Review:**

Summary:
The authors improve sparse for recurrent neural networks by developing a greedy redistribution rule for gates and adapting the ASGD optimizer for sparse networks. The work provides good results and a rich analysis of their and related methods.

Strong points:
- very rigorous experimental setup and analysis
- Solid evidence for many new insights into some sparse training phenomena. The work provides broadens our understanding of sparse training.

Weak points:
- Some might complain that RNNs are outdated. I see this only as a minor weak point. Indeed, RNNs are not much used anymore, but many of the insights the paper provides are quite universal.
- The fixed FLOPS only seems to be a by-product of the algorithm and particular network structure but not necessarily an algorithmic contribution. This makes the paper a bit confusing.

Recommendation (short):
This is a very solid paper with exemplary experimentation and analysis. It provides many unique insights that are very valuable for anyone who wants to work in the field of sparse training. I recommend accepting this paper.

Recommendation (long):
I think this paper is one of these papers, which is a very solid all-around. The authors invested quite a bit of time in creating rigorous experimental setups that test hypotheses. In particular, I like the graph analysis of sparse connective between networks. Findings of different initialization schemes and performance of other sparse training methods are precious and make the overall literature on sparse training robust. I can see that this paper may seem a bit boring and less impactful to some reviewers, but good science like this is not about being exciting but about providing rigorous results for a small problem. This paper does exactly that. I think any good conference should encourage good science by accepting papers like this one.

Comments for authors:
Solid work. Here some additional comments and questions.
- Please feed your paper through a grammar/spellchecker. There are multiple errors which make the paper hard to read in some sections
- It is not entirely clear why ASGD is needed for good performance. Can you elaborate, please?
- Do you have any idea how does ER initialization relates to eigenvalues of recurrent matrices? If you can make a connection here, it would be a quite insightful addition to the paper since the top eigenvalue of the recurrent matrix determines the overall long-term behavior of the recurrent matrix and is known to influence behavior.
- I would drop the fixed FLOPS contribution and focus on the other parts of the paper. You have more than enough contributions, and the space is better devoted to making the other contributions as clear as possible.
- The cell weight redistribution algorithm description is unclear. A weight cannot have "more parameters, I think you mean to say gate-neurons with large magnitude weights gain more parameters over time.
- The sparse topology algorithm: Is the correlation between weights computed overall test set outputs between two networks/weights?
- Figure 3, unclear. What does Figure 3 (left) show exactly? It is unclear what random initialization means: different sparsity patterns, different weight values, or both? What does the seed do here? Does it affect sparsity pattern, data order, weight values, etc.?

---

> ### Author Response · Authors · 2020-11-20
> **AnonReviewer3 Author Response**
>
> We would like to thank you for providing a fascinating perspective on our work. We are glad that our paper can provide some sparse training insights to you.
>
> Our response to your comments and questions:
> - **Please feed your paper through a grammar/spellchecker. There are multiple errors which make the paper hard to read in some sections.** We have carefully polished our paper and updated the new version. Also, we will continue this process in the following updates.
> - **It is not entirely clear why ASGD is needed for good performance. Can you elaborate, please?** There are many works that provide theoretical analysis of convergence for averaged SGD. Polyak and Juditsky (1992) [1] proved that, among stochastic gradient algorithms, averaged SGD achieves the best possible convergence rate, even better than methods that use second-order information. Further, Mandt et al. [2] showed that iterative averaged SGD can obtain a lower-variance optimum estimator of the loss function. Despite its theoretical appeal, the unclear guidelines of the averaging trigger $T$ and the learning rate scheduler make it difficult to be used to train neural networks. Recently, Merity et al. [3] proposed NT-ASGD, wherein a non-monotonical condition is used to trigger the averaging process, yielding better performance than SGD in RNNs. Therefore, it is reasonable to use SNT-ASGD to improve the performance of sparse training.
> - **Do you have any idea how does ER initialization relates to eigenvalues of recurrent matrices? If you can make a connection here, it would be a quite insightful addition to the paper since the top eigenvalue of the recurrent matrix determines the overall long-term behavior of the recurrent matrix and is known to influence behavior.** Early work has shown that initializing the recurrent matrix as identity or orthogonal may improve long-term dependencies of sequences [4,5]. As you suggested, we calculate the top-5 eigenvalues of the recurrent matrix, as shown below. It seems the eigenvalues of ER distribution is a bit higher than the eigenvalues of uniform distribution. This phenomenon is very interesting and worthy of further study. We don't want to just give a superficial conclusion, as we believe the situation of sparse RNNs is more complex than the one of dense RNNs. We prefer to fully study it as a future research work.
> | Overall Density | Sparse distribution |     Top-5 Eigenvalues    | Density of RNN layers |
> |:---------------:|:-------------------:|:------------------------:|:---------------------:|
> |       0.05      |          ER         | 0.14,0.14,0.14,0.13,0.13 |          0.06         |
> |       0.05      |       Uniform       | 0.13,0.13,0.12,0.12,0.12 |          0.05         |
> |       0.33      |          ER         | 0.37,0.37,0.36,0.36,0.35 |          0.34         |
> |       0.33      |       Uniform       | 0.34,0.33,0.33,0.32,0.32 |          0.33         |
> - **I would drop the fixed FLOPS contribution and focus on the other parts of the paper. You have more than enough contributions, and the space is better devoted to making the other contributions as clear as possible.** Agree! We have made the suggested change.
> - **The cell weight redistribution algorithm description is unclear. A weight cannot have "more parameters, I think you mean to say gate-neurons with large magnitude weights gain more parameters over time.** Yes, you are right. We actually mean the gates with more large-magnitude weights gain more weights. We have made the suggested change in the new version.
> - **The sparse topology algorithm: Is the correlation between weights computed overall test set outputs between two networks/weights?** Yes, the correlation is calculated with the whole test set.
> - **Figure 3, unclear. What does Figure 3 (left) show exactly? It is unclear what random initialization means: different sparsity patterns, different weight values, or both? What does the seed do here? Does it affect sparsity pattern, data order, weight values, etc.?** Figure 3-left shows how the sparse topology evolves when optimized by Selfish-RNN. We just trained a sparse network with a random seed and check how its sparse topology changes during training by comparing the sparse connectivity at the $5^{th}$ epoch with the sparse connectivity at the following epochs.
>
> [1] [Acceleration of Stochastic Approximation by Averaging](https://epubs.siam.org/doi/pdf/10.1137/0330046)
> [2] [Stochastic Gradient Descent as Approximate Bayesian Inference](https://arxiv.org/abs/1704.04289)
> [3] [Regularizing and Optimizing LSTM Language Models](https://arxiv.org/abs/1708.02182)
> [4] [Recurrent Orthogonal Networks and Long-Memory Tasks](https://arxiv.org/abs/1602.06662)
> [5] [A Simple Way to Initialize Recurrent Networks of Rectified Linear Units](https://arxiv.org/abs/1504.00941)

---

### Official Review · AnonReviewer4 · 2020-10-26
**An interesting paper that proposes an approach to train sparse recurrent models, and a sparse variant of the NT-ASGD.**

**Rating:** 6
**Confidence:** 3

**Review:**

In this paper, the authors propose an approach to train sparse recurrent models, and a sparse variant of the NT-ASGD. The proposed method mixes some interesting novel methodologies and achieves interesting empirical results on Penn Treebank and WikiText-2 language modeling tasks.
In general, the paper is well written and interesting, but in section 3 many explanations about the rationale behind some architectural choices of the selfish-RNN methodology are only partially explained, and sometimes they are just related to empirical results (e.g. in the cell weight redistribution). To me, a more theoretical explanation would significantly improve the manuscript readability.
In section 4 many different approaches were considered. But there are a few points that are not clear. The authors report the results of a “small" dense network, but no information about this model is reported in the text.
Reading the results reported in table 5 of the appendix, I found it interesting that the performance of the DSR improves significantly by using SNT-ASGD instead of Adam  (it outperforms the Selfish-RNN). This table shows how much the optimizer influences model performance.  Even if the ablation study reported in appendix A highlights the benefits of the SNT-ASGD, the results reported in table 5 show that the impact of this component is even more important than the selfish-RNN. Honestly, I think that is fairer to compare all the methods using the same optimization algorithm, therefore my suggestion is to move this table in the main paper and extend the analysis of these results.
Reading the manuscript it is not clear how the hyper-parameters considered in the experimental campaigns have been chosen. By reading the first part of section 4.1 seems like parameters like the removing rate or the number of epochs are set without performing any validation on them. Even in appendix D, hyper-parameters (e.g. the learning rate, or the batch size) used to test the RHM are just listed. The authors should insert a more extensive explanation about how the hyper-parameters various models/approaches considered in the comparison have been validated. To perform a fair comparison the hyper-parameters of each model should be chosen according to its performance on the validation set.
In this regard, it is important also to highlight how the hyper-parameters are chosen because some SOTA models achieved better results. For instance on the Penn Treebank dataset in “On The State Of The Art Of Evaluation In Neural Language Models”, Melis et al. report perplexities on the test set of 59.7.
exploiting better the research space. The reported results in the paper (and in Appendix L) show the benefits of using this approach, but honestly, to me, it is not clear if it helps in exploring the state space. In general, it is not clear what is the reason why the model benefits from using the random growth approach. Moreover, in “Sparse evolutionary deep learning with over one million artificial neurons on commodity hardware” the gradient guided growth strategy outperforms the other sparse training approaches considered in the paper, even in the RNN case. Therefore a more extended evaluation/discussion of this point is required.
Another recently proposed approach that uses sparsity in recurrent models is defined in “Intrinsically Sparse Long Short-Term Memory Networks” by Liu et al. the author should compare this approach with the selfish-LSTM.

---

> ### Author Response · Authors · 2020-11-20
> **AnonReviewer4 Author Response (2/2)**
>
> Continued Response ...
> - **In this regard, it is important also to highlight how the hyper-parameters are chosen because some SOTA models achieved better results.**
>   - For hyperparameters introduced by Selfish-RNN: sparsity and the initial pruning rate, we have provided an analysis in the Appendix F and Appendix G, respectively. We find that Selfish Stacked LSTMs, RHNs, ON-LSTM, and AWD-LSTM-MoS need around 25%, 40%, 45%, and 40% parameters to match the performance of their dense counterparts, respectively. And we can clearly see that our method is very robust to the choice of the initial pruning rates.
>   - For the original hyperparameters of RNNs, we clarify that we didn’t apply any hyperparameter tuning techniques used in “On the State of the Art of Evaluation in Neural Language Models”. Instead, we use exactly the same hyper-parameters and regularization as the dense ON-LSTM and AWD-LSTM-MoS. Further, we extended similar settings to stacked LSTM and RHN, since there are no existing implementations of these two models trained with NT-ASGD. No need of finetuning the original hyperparameters of the dense model is another advantage of our method. Moreover, for a fair comparison, the hyperparameters of all DST methods are the same.
> - **Moreover, in “Sparse evolutionary deep learning with over one million artificial neurons on commodity hardware” the gradient guided growth strategy outperforms the other sparse training approaches considered in the paper, even in the RNN case. Therefore a more extended evaluation/discussion of this point is required. Another recently proposed approach that uses sparsity in recurrent models is defined in “Intrinsically Sparse Long Short-Term Memory Networks” by Liu et al. the author should compare this approach with the selfish-LSTM.**
>   - We carefully read the paper “Sparse evolutionary deep learning with over one million artificial neurons on commodity hardware” and didn’t find anything about gradient-based growth. This paper just provides more motivation to our method as it develops for the first time an independent software framework to train very large truly sparse multilayer perceptrons trained with SET. We have cited it in Appendix N. We suppose that you mean this paper “Rigging the Lottery: Making All Tickets Winners”. It shows that even in the RNN case, the gradient-based approaches (RigL and SNFS) outperform the other sparse training approaches. There is more than just random or gradient growth in the DST family of methods, and those results are in line with ours. As detailed in the Table above, we can see that based on the optimizer the DST methods reach a different level of performance.
>   - For the paper “Intrinsically Sparse Long Short-Term Memory Networks”, we found that it is a straightforward extension of applying SET to RNN. We have discussed it properly in the new version of our paper.

---

> ### Author Response · Authors · 2020-11-20
> **AnonReviewer4 Author Response (1/2)**
>
> We would like to thank you for giving us constructive feedback. We provide the following responses to your concerns.
> * **The authors report the results of a “small" dense network, but no information about this model is reported in the text.** The small dense networks refer to dense networks with the same number of parameters as Selfish-RNN, this automatically yielding fewer hidden units. The small dense stacked LSTM has 700 hidden units and the small dense RHN has 510 hidden units.
> * **I think that is fairer to compare all the methods using the same optimization algorithm, therefore my suggestion is to move this table in the main paper and extend the analysis of these results.**
> * **In general, it is not clear what is the reason why the model benefits from using the random growth approach.** We address these two concerns together. We fully agree that it would be fairer to compare different dynamic sparse training (DST) methods with the same optimizer. To understand better the effect of different optimizers on different DST methods, we have extended Table 5 with an extra group trained with momentum SGD and we have moved it in the main paper as Table 3. We use a learning rate of 2 decreased by a factor of 1.33 once the validation loss fails to decrease. Similar strategies are also used in the original stacked LSTM and RHN papers. For convenience, we share the results at the bottom of this response. We can see that different optimizers have different influences on the DST methods. For example, DSR achieves good performance when trained with momentum SGD and SNT-ASGD, but it has the worst performance when trained with Adam. And the performance of gradient-based growth is also greatly affected by the optimizer. The perplexity gaps between RigL and Selfish-RNN when trained with momentum SGD (1.12) and Adam (2.76) are much smaller than the one trained with SNT-ASGD (4.25). However, the performance of Selfish-RNN is quite robust due to its simple scheme and clear backward pass. Advanced strategies such as across-layer weight redistribution used in DSR, SNFS, and gradient-based weight regrowth used in RigL, SNFS heavily depend on optimizers. They might work decently for some optimization methods, but may not work for others.
> | Models                 | #Param | Val   | Test  |
> |------------------------|--------|-------|-------|
> | DSR (Adam)             | 21.8M  | 89.95 | 88.16 |
> | SET(Adam)              | 21.8M  | 88.31 | 86.28 |
> | SNFS(Adam)             | 21.8M  | 88.39 | 85.61 |
> | RigL(Adam)             | 21.8M  | 87.30 | 85.49 |
> | Selfish-RNN(Adam)      | 21.8M  | **85.70** | **82.85** |
> | SNFS (Momentum)        | 21.8M  | 90.09 | 87.98 |
> | SET (Momentum)         | 21.8M  | 85.73 | 82.52 |
> | RigL (Momentum)        | 21.8M  | 84.78 | 80.81 |
> | DSR (Momentum)         | 21.8M  | 82.89 | 80.09 |
> | Selfish-RNN (Momentum) | 21.8M  | **82.48** | **79.69** |
> | SNFS (SNT-ASGD)        | 21.8M  | 82.11 | 79.50 |
> | RigL (SNT-ASGD)        | 21.8M  | 78.31 | 75.90 |
> | SET (SNT-ASGD)         | 21.8M  | 76.78 | 74.84 |
> | Selfish-RNN (SNT-ASGD) | 21.8M  | 73.76 | 71.65 |
> | DSR (SNT-ASGD)         | 21.8M  | **72.30** | **70.76** |

---

### Official Review · AnonReviewer2 · 2020-10-27
**minor technical novelities that lead to improved performance over state of the art**

**Rating:** 7
**Confidence:** 3

**Review:**

The paper claims that the previous sparse training methods mainly focus on MLP and CNN, and fail to perform very well in RNNs. Hence, the authors proposed an approach to train sparse RNNs with a fixed FLOPs budget.
The proposed technique is based on defining a mask matrix $M$ and refining it during training. It is initialized randomly to have the desired sparsity level $S$. After each training epoch, a fraction $p$ of the weights with the smallest magnitude is removed, i.e., those locations are zeroed in the mask M.
Next, the same amount of parameters are randomly added to M again.
Moreover, a variant of the averaged stochastic gradient optimizer (SNT-ASGD) is developed for the training of sparse RNN to account for the effect of weight masks during training.
They showed that in practice, the requirements for efficient sparse training of RNNs are different than CNN and MLP.

Strengths:
By adding some refinements and tweaks to the existing techniques (masking for sparse training and adapting the NT-ASGD), the authors were able to achieve good performance to train sparse RNNs. The paper has a rather extensive set of simulations and experimental setups to analyze the best setup which yields good sparse training, e.g., comparing uniform vs ER distribution for masks, sensitivity to hyperparameters, ... Moreover, they have considered a fairly diverse set of RNN architectures to evaluate their method.

Weaknesses and questions:
Compared to the existing methods, the technical novelty of the paper is minor. It can be seen as some tweaks and improvements to the existing ones (although I admit that those changes are essential for the method to work for RNN.).
What is special about the method that makes it specific to RNN? In other words, is it possible to use the same method for sparse training of MLP and CNN?
A minor issue with the paper is the FLOPS analysis the authors used. Effectively, they use the sparsity of the parameters as a measure of FLOPS, not the actual FLOPS that might depend on the sparsity structure, HW, or software implementation. It would be a good idea to directly mention and use total sparsity, instead of FLOPS which can mislead the readers.

Some parts of the method are not clear enough, e.g.,
1. In the paper, it is stated that "magnitude weight removal" is applied to non-RNN layers. Do the authors mean that for the parameters of RNN, this step is skipped?
2. In "cell weight redistribution", it is suggested that the "magnitude weight removal" is applied to the whole set of RNN parameters $\{\theta_1, \ldots, \theta_t\}$. However, in "random weight growth", it is mentioned that the same number of weights is grown immediately after weight removal, i.e., $R$ and $P$ have the same number of 1's. So, does it mean that the number of 1's in mask $M_i$ for each weight $\theta_i$ ($1\leq i \leq t$) remains fixed S during training?
3. Another aspect of training that is unclear for me is the parameters that are updated. Is $\theta$ updated during training or only $\theta_s$ is updated? As a result, if a weight is removed in one epoch and its value at the time of removal was $\alpha$, and later regrown at another epoch, is its initial value set to 0 or started from its previous value before "weight removal", i.e. $\alpha$?
4. Did the authors add any regularizer (e.g., $\ell_1$) to the training loss to improve sparsity in their experiments?

---

> ### Author Response · Authors · 2020-11-20
> **AnonReviewer2 Author Response**
>
> We thank the reviewer for the positive feedback and we would like to answer your questions in detail below:
> - **What is special about the method that makes it specific to RNN? In other words, is it possible to use the same method for sparse training of MLP and CNN?** The special components of our method are (1) the proposed SNT-ASGD which overcomes the over-sparsified problem of the standard NT-ASGD in sparse RNNs, and (2) cell weight redistribution, allowing RNN layers to have a non-uniform distribution. As far as we understand, it is possible to apply these techniques to train sparse CNN and MLP, but it is not in the main scope of our paper. We explain them as follows.
>   - NT-ASGD was proposed in [1], wherein a non-monotonical condition is used to trigger the averaging process, yielding better performance than SGD in RNNs. Therefore, it is reasonable to use SNT-ASGD to improve the performance of sparse RNN training. However, the effectiveness of SNT-ASGD on CNN and MLP still needs to be evaluated. Two additional hyperparameters, the logging interval $L$ and non-monotone interval $n$, might need to be finetuned. In addition to NT-ASGD, other alternatives may be more suitable to train other types of sparse neural networks. For instance, stochastic weight averaging (SWA) [2] can be a good example of utilizing weight averaging to achieve better generalization performance over the conventional training with modern CNN architectures.
>   - Cell weight redistribution can not be directly applied to other architectures. One possible way may be applying a similar strategy to the filters of each CNN layer, but this is just a hypothesis and would have to be evaluated further.
> * **A minor issue with the paper is the FLOPS analysis the authors used.** We agree with your comment on the FLOPs analysis. We used the FLOP approximation, as typically done also in other papers on sparse neural networks (e.g., [3]), rather than the actual FLOPs because the latter one depends on the specific hardware or software implementation. Also, we would need a truly sparse implementation of our proposed method which is not easy to have as all deep learning hardware and software libraries do not properly support yet operations with sparse connectivity matrices. With our choice, we just want to highlight that different DST methods have different sparse connectivity distributions, which lead to very different computational costs.
> * **In the paper, it is stated that "magnitude weight removal" is applied to non-RNN layers. Do the authors mean that for the parameters of RNN, this step is skipped?** For non-RNN layers, we use magnitude weight removal followed by random weight growth to update the sparse connectivity. After the weight removal, the same number of weights (parameters) are randomly grown (added) to each non-RNN layer. Therefore, for non-RNN layers, the number of non-zero weights is fixed. For RNN layers, we only use cell weight redistribution to update their sparse connectivity. We first apply "magnitude weight removal" to the whole set of RNN weight tensors $\theta_1,...,\theta_t$, so that different weight tensor $\theta_t$ will have different pruning rates. Then, new weights are grown to all cell weight tensors $\theta_1,...,\theta_t$ uniformly. By doing this, the total number of weights in each RNN layer is fixed, but the number of weights in each cell weight tensor is dynamic.
> * **Another aspect of training that is unclear for me is the parameters that are updated.**  Since we use masks to simulate the sparse training, in our PyTorch implementation, the entire weights $\theta$ will be updated during the backward pass. We further use Eq. (1): $\theta_s = \theta \odot M$ to enforce the sparse structure before the forward pass and after the backward pass, so that the zero-valued weights will not contribute to the loss. Moreover, the newly activated weights are initialized to zero.
> * **Did the authors add any regularizer (e.g., ℓ1) to the training loss to improve sparsity in their experiments?** We don’t use any regularizer to improve the sparsity. Techniques that leverage the $l_0$ or $l_1$ regularizer to induce sparsity are categorized as a dense-to-sparse approach and are motivated for inference efficiency. A dense model has to be trained for some time, and the sparse model is learned gradually during the training. In contrast, our method is motivated for training efficiency, starting with a sparse model. Of course, the use of such regularizers in sparse training may help in obtaining even a higher sparsity level. We believe that this can be an interesting future work direction, but we prefer to let it out of this paper.
>
> [1] [Regularizing and Optimizing LSTM Language Models](https://arxiv.org/abs/1708.02182)
> [2] [Averaging weights leads to wider optima and better generalization](https://arxiv.org/abs/1803.05407)
> [3] [Rigging the lottery: Making all tickets winners](https://arxiv.org/abs/1911.11134)

---

### Author Response · Authors · 2020-11-21
**General response to all reviewers**

We thank the reviewers for the evaluation of our manuscript and for their valuable comments.  We are grateful for the constructive feedback, which helps us substantially improve our paper. We briefly summarise the main changes we have made during the rebuttal.
- To understand better the effect of different optimizers on different DST methods, we have moved Table 5 in the main paper as Table 3 with an extended group of experiments trained with momentum SGD.
- We have carefully polished our paper and updated the new version. Also, we will continue this process in the following updates.
- As suggested by AnonReviewer2&3, we have dropped the fixed FLOPs contribution.

---

### Decision · Program_Chairs · 2021-01-07
**Final Decision**

**Decision:**

Reject

**Comment:**

The authors introduce an approach to train sparse RNNs with a fixed parameter count. During training, they allow RNN layers to have a non-uniform redistribution across cell weights for a better regularization.They also introduce a variant of the averaged stochastic gradient optimizer, which improves the performance of all sparse training methods for RNNs. They achieve state-of-the-art sparse training results on Penn Treebank and Wikitext-2.

The method achieves very good performance on sparse RNNs for challenging tasks. The paper is well written and provides solid analysis with new insights into sparse network models. Most reviewers believe it is a very solid paper.

However, the technical novelty of the paper is limited. It can be seen as some tweaks and improvements of existing techniques, which seem to work very well. Since the number of papers that can be accepted is very limited, and since technical novelty is an essential criterion for published papers at ICLR, I propose rejection.